# Effect of High Static Magnetic Fields on Biological Activities and Iron Metabolism in MLO-Y4 Osteocyte-like Cells

**DOI:** 10.3390/cells10123519

**Published:** 2021-12-13

**Authors:** Jiancheng Yang, Gejing Zhang, Qingmei Li, Qinghua Tang, Yan Feng, Peng Shang, Yuhong Zeng

**Affiliations:** 1Department of Osteoporosis, Honghui Hospital, Xi’an Jiaotong University, Xi’an 710054, China; yangjiancheng@nwpu.edu.cn (J.Y.); liqingmei09@163.com (Q.L.); gshua691994@163.com (Q.T.); fengyan120119@163.com (Y.F.); 2Key Laboratory for Space Bioscience and Biotechnology, School of Life Sciences, Northwestern Polytechnical University, Xi’an 710072, China; 2017263324@mail.nwpu.edu.cn; 3Research & Development Institute in Shenzhen, Northwestern Polytechnical University in Shenzhen, Shenzhen 518057, China

**Keywords:** high static magnetic fields, osteocytes, iron metabolism, cell viability, cytoskeleton, apoptosis

## Abstract

There are numerous studies that investigate the effects of static magnetic fields (SMFs) on osteoblasts and osteoclasts. However, although osteocytes are the most abundant cell type in bone tissue, there are few studies on the biological effects of osteocytes under magnetic fields. Iron is a necessary microelement that is involved in numerous life activities in cells. Studies have shown that high static magnetic fields (HiSMF) can regulate cellular iron metabolism. To illustrate the effect of HiSMF on activities of osteocytes, and whether iron is involved in this process, HiSMF of 16 tesla (T) was used, and the changes in cellular morphology, cytoskeleton, function-related protein expression, secretion of various cytokines, and iron metabolism in osteocytes under HiSMF were studied. In addition, the biological effects of HiSMF combined with iron preparation and iron chelator on osteocytes were also investigated. The results showed that HiSMF promoted cellular viability, decreased apoptosis, increased the fractal dimension of the cytoskeleton, altered the secretion of cytokines, and increased iron levels in osteocytes. Moreover, it was found that the biological effects of osteocytes under HiSMF are attenuated or enhanced by treatment with a certain concentration of iron. These data suggest that HiSMF-regulated cellular iron metabolism may be involved in altering the biological effects of osteocytes under HiSMF exposure.

## 1. Introduction

Because of the continuous advances of industrial and medical technology, stronger magnetic fields are needed, which means that humans will be exposed to static magnetic fields (SMFs) with higher magnetic field strength. Magnetic resonance imaging (MRI) is the most common high static magnetic fields (HiSMF) to which humans are exposed. Currently, the magnetic field intensity of MRI commonly used in medical institutions is 1.5–3 T. MRIs at 7 T are already authorized by the Food and Drug Administration (FDA) for clinical diagnostics [1]. Recently, the strongest MRI machine, with a 10.5 T magnetic field intensity, was established at the University of Minnesota to scan the human body [2]. In addition, scientists have built a 21.2 T MRI to scan the brain of rats [3]. Thus, the biological effects of organisms under HiSMF should be revealed. However, the limited reports thus far have focused on studies of the nervous system, cardiovascular system, and hematologic system. Studies on the effects of HiSMF on the skeletal system are scarce, especially those related to osteocytes.

Osteocytes, accounting for more than 90% of all cells in bone, are the most abundant cells in bone tissue. There are few reports on the effects of HiSMF on osteocytes, and only some preliminary studies on osteocytes have been previously conducted by our laboratory using a 16 T HiSMF generated by a large gradient superconducting magnet. MLO-Y4 cells, an osteocyte-like cell, from the long bone of a transgenic mouse expressing SV40 large T-antigen [4], was exposed to the HiSMF environment for 48 h, and the osteocytic gene expression profile was analyzed by a gene microarray, revealing that the expression of energy metabolism-related genes was affected by HiSMF in osteocytes [5]. Forty-nine genes were significantly altered (four upregulated and 45 downregulated) in MLO-Y4 cells after short (3 h) exposure to HiSMF [6]. It was found that HiSMF had no significant effect on the cell viability of osteocytes, indicating that HiSMF had no lethal effect on osteocytes [7]. Nevertheless, the effects of HiSMF on osteocytes have not been systematically studied, and their effects on osteocyte function and related molecular expressions as well as various secreted factors are not clear. In addition, the influences of HiSMF on the expression of functional molecules and the secretion of various cytokines are unclear.

Iron is a microelement for all living organisms, enrolled in many cellular activities, including DNA replication and cellular respiration [8,9]. However, excess iron catalyzes the Fenton reaction in cells, which produces large amounts of reactive oxygen species (ROS) that are toxic to cells [10]. Therefore, both iron overload and iron deficiency can exert a negative impact on the basic life activities of the cell. There is increasing evidence that excessive iron promotes osteoclastic formation and bone resorption [11,12], while inhibiting osteoblastic differentiation and bone formation [13,14,15]. In contrast, iron chelation can inhibit osteoclastic differentiation and enhance the biological activity of osteoblasts [16,17]. However, as the most abundant cell in bone tissue, few studies have reported how iron affects the function of osteocytes.

Few studies have reported the influence of magnetic fields on iron metabolism, although iron and magnetism have a natural connection. Recently, we found that HiSMF of 16 T facilitated osteoblast proliferation with elevated iron levels and triggered osteogenic mineralization with a decrease in iron levels, which was modulated via blocking iron acquisition and elevating iron export [18]. Moreover, we demonstrated that HiSMF markedly restrained osteoclastic formation and decreased cellular iron content during the differentiation of osteoclasts [19]. Mechanistically, HiSMF significantly reduced the expression of osteoclastic differentiation-associated genes and prevented iron influx and iron sequestration-related protein expression. Therefore, cellular iron metabolism can be modulated by HiSMF. However, whether HiSMF can influence the function of osteocytes by influencing cellular iron metabolism requires further study.

The purpose of this study was to explore the biological effects of HiSMF on osteocytes and the regulation of cellular iron metabolism in osteocytes, and to elucidate the correlation between them. The changes in cellular morphology, the cytoskeleton, function-related protein expression and secretion of various cytokines will be examined in osteocytes under a 16 T HiSMF. The changes in iron metabolism in osteocytes will also be evaluated. Furthermore, the effects of iron overload and iron deficiency on osteocytes will be investigated by supplementing exogenous iron preparation and iron chelating agents under a geomagnetic field (GMF) and HiSMF, respectively.

## 2. Materials and Methods

### 2.1. Magnetic Field Exposure System

HiSMF of 16 T was produced by a superconducting magnet (JMTA-16T50MF), which was built by Japan Superconductor Technology (JASTEC, Kobe, Honshu, Japan). As we described previously, an experimental system for cellular culture in the superconducting magnet was developed, including a gas regulation system, a temperature control system, and an object stage [20]. Briefly, the concentration of CO_2_ was 5% adjusted by air flow meter, and the temperature was maintained at 37 °C by a homothermal water-bath pump (Ouhai Apparatus, Cangzhou, China) in the cylinder bore of the magnet with 50 mm diameter (Figure 1A). The object stage contains different platforms that correspond to different magnetic field intensities and different magnetic field gradients (Figure 1B–D). In this study, the cells were placed in the position of 16 T with 0 T^2^/m.

### 2.2. Cell Culture

The Osteocyte-like cell line, MLO-Y4, was gifted by Prof. Jean X. Jiang (University of Texas Health Science Center, San Antonio, TX, USA) and used in this study. A collagen-coated petri dish with α-MEM medium (Gibco, Grand Island, NY, USA), which was supplemented with 5% calf serum (Gibco), 5% fetal bovine serum (Gibco), 1% penicillin–streptomycin, and 2 mM L-glutamine was used to culture osteocytes. The cells were incubated in 37 °C incubator with humid atmosphere with 5% CO_2_. The cultural supernatant was replaced every 2 days.

### 2.3. Cell Viability Assay

MLO-Y4 cells at the logarithmic phase were inoculated in the microtiter wells of 96-well plates at 1000 cells/well, and the cells were cultivated under normal conditions for 12 h. Following, they were incubated under GMF and HiSMF for 24 and 48 h or supplemented with different concentrations of desferrioxamine (DFO) or ferric ammonium citrate (FAC) in the medium and continued to incubate for 48h under HiSMF and GMF. Then, cell viability of MLO-Y4 cells was tested by using the Cell Counting Kit 8 kit (Beyotime Biotechnology, Shanghai, China). Briefly, the culture supernatant was discarded, and 100 μL of medium with 10% volume of CCK-8 was added and incubated at 37 °C for 3 h. A Synergy HT multimode microplate reader (BioTek, Winooski, VT, USA) was used to examine optical density (OD) value at 450 nm.

### 2.4. Cell Morphology Observation

Seed MLO-Y4 cells were loaded onto 18 mm petri dishes at the logarithmic phase and were allowed to attach after 12 h, and then they were immediately placed in 16 T HiSMF for 48 h or supplemented with different concentrations of DFO or FAC in the medium and exposed to HiSMF and GMF for 48 h. After treatment, 4% paraformaldehyde was added to fix for 20 min, purified water was used to wash once, 0.1% crystal violet solution was added to stain for 30 min, which was washed cleanly using ultrapure water. Then, the Nikon Eclipse 80i microscope (Nikon, Tokyo, Japan) was used to observe and photograph the cells. One hundred cells were selected, and each index of cell morphology was counted by using Image-Pro Plus 6.0 software (Media Cybernetics, Houston, TX, USA).

### 2.5. Cell Apoptosis Assay

MLO-Y4 cells were seeded on a slide at the logarithmic phase and exposed to HiSMF for 48 h or added to different concentrations of DFO or FAC in the medium and exposed to HiSMF and GMF for 48 h; the supernatant was discarded, fixed in 10% formaldehyde solution for 20 min, and phosphate buffer saline (PBS) was used to wash three times. Hoechst 33,258 staining solution (Beyotime) was added and stained at indoor temperature for 5 min. PBS was used to wash twice and the liquid was blotted using filter paper. The slide was sealed using glycerol, avoiding air bubbles as much as possible. The Nikon Eclipse 80i microscope (Tokyo, Honshu, Japan) was used to observe and photograph the cell nucleus with blue-fluorescence. In the event of apoptosis, the chromatin was solidified and the nuclei were densely stained or fragmented with whitish color, while normal cells had normal blue nuclei. The total amount of cell nuclei and the number of apoptotic nuclei in each sight were counted using Image-Pro Plus software. Three sights per sample were counted. The percentage of apoptotic cells was determined by calculating the ratio of the amount of apoptotic nucleus to the total number of cell nuclei.

### 2.6. Cytoskeleton Staining

MLO-Y4 cells were seeded on a slide in 18 mm petri dishes at a density of 2 × 10^4^ cells/dish. After attaching for 12 h, they were exposed to HiSMF for 48 h, or added to different levels of DFO or FAC in the medium and incubated under HiSMF and GMF conditions for 48 h. The supernatant was removed, and PBS was used to wash the cells once, and then it was fixed with 4% paraformaldehyde for 20 min. Next, 0.5% Triton X-100 (Aladdin, Shanghai, China) solution was used to permeabilize the cells for 10 min at room temperature, was blocked with PBS containing 2% BSA for 10 min. Rhodamine-labeled ghost cyclic peptide and tubulin monoclonal antibody were added, and it incubated overnight at 4 °C, protected from light. Triton X-100 solution (0.1%) was used to wash 3 times, and the cells were incubated with FITC-labeled secondary antibody and DAPI for 1 h and were protected from light, and 0.1% Triton X-100 solution was used to wash the cells 3 times. The slides with cells attached were sealed on the microslide using anti-fluorescence quencher, avoiding air bubbles as much as possible. Then, the cytoskeleton was observed and photographed using the Nikon Eclipse 80i microscope. The fractal dimension of microfilaments and microtubules was evaluated using ImageJ software.

### 2.7. Soluble Cytokines Assay

Cell culture supernatants of MLO-Y4 cells were collected after exposure to HiSMF for 48 h, or treatment with different levels of DFO or FAC under HiSMF and GMF conditions for 48 h. The concentrations of alkaline phosphatase (ALP), prostaglandin E2 (PGE2), nitric oxide (NO), and Nuclear Factor κB Receptor Activator Ligand (RANKL) in the supernatants were examined. ALP concentrations in the supernatant were assayed by using a commercial mouse ALP kit (Beyotime). The levels of PGE2 and RANKL in the supernatant were measured using the Mouse PGE2 ELISA Kit and the Mouse RANKL ELISA Kit (Shanghai Jianglai Biological Technology, China), respectively. The concentration of NO in the supernatant was detected by using a commercial nitric oxide assay kit (Beyotime).

### 2.8. Cellular Iron Content Assay

As we described previously, iron content in osteocytes was determined by atomic absorption spectrometry (AAS; Analytik Jena, Jena, Germany) [21]. MLO-Y4 osteocyte-like cells were exposed to HiSMF for 48 h, and then normal saline was used to wash the cells three times and was dissolved in 65% HNO_3_ at 70 °C for 2 h. The lysate was deliquated in 10 mL ultrapure water. Cellular iron levels were tested by graphite furnace atomic absorption spectroscopy and were normalized by the total protein content or the total amount of cells.

### 2.9. Protein Expression Assay

MLO-Y4 cells were subjected to HiSMF for 48 h; the RIPA Lysis Buffer (Beyotime) containing 1-mM Phenylmethanesulfonyl fluoride (PMSF; Beyotime) was used to lyse the cells. A commercial BCA Protein Assay Kit (Beyotime) was used to examine the protein content. The proteins were loaded into the SDS-PAGE gels and were separated by electrophoresis, and then transferred onto the nitrocellulose (NC) membrane. The NC membranes were blocked with 5% nonfat-dried milk and were then incubated in the specific primary antibody, including anti-GAPDH (Proteintech, Rosemont, IL, USA), anti-Connexin 43 (Cx43；BioWorld, Nanjing, Jiangsu, China), anti-Sclerostin (Abcam, Cambridge, Cambridgeshire, UK), anti-Ferritin Heavy Chain (FTH1; Cell Signaling Technology, Boston, MA, USA), anti-divalent metal transporter 1 (DMT1; Abcam), and anti-ferroportin 1 (FPN1/SLC40A1; Abcam) overnight at 4 °C. After using tris buffered saline tween (TBST) to wash three times, NC membranes were incubated by the species-specific secondary antibody conjugated to horseradish peroxidase (Kangwei Century, Beijing, China) for 1 h at indoor temperature. A ECL Plus Western Blotting Detection System (Tanon, Shanghai, China) was used to image the immunoreactive bands. The grayscale bands were determined by ImageJ 1.8.0 software (NIH, Bethesda, MD, USA).

### 2.10. Statistical Analysis

All experimental data were expressed as mean ± SD, and statistical analysis was processed using GraphPad Prism 6.0 software (GraphPad Software, La Jolla, CA, USA). The unpaired t test was used to analyze the differences between GMF and HiSMF. Two-way ANOVA with Sidak’s multiple comparison method was used to evaluate the differences between GMF and HiSMF after increasing or decreasing iron. Differences with *p* < 0.05 were considered statistically significant.

## 3. Results

### 3.1. The Effect of HiSMF on Cell Growth

To examine the cell growth of osteocytes under HiSMF, cell viability, apoptosis and morphology of MLO-Y4 cells were determined. The results showed that treatment of osteocytes with HiSMF for 48 h significantly promoted cell viability (Figure 2A) and inhibited osteocyte apoptosis (Figure 2B,C). No significant influence on cell morphology and area (Figure 2D,E), and on the number and length of processes (Figure 2F), was found. These results indicate that HiSMF contributes to the growth of osteocytes and has no effect on the morphology of osteocytes.

### 3.2. The Effect of HiSMF on Cytoskeleton

The cytoskeleton plays a key role in maintaining cell morphology, withstanding external forces, and sustaining orderliness of the internal cell structure. The cytoskeleton, including microfilaments (Figure 3A) and microtubules (Figure 3B), was evaluated in osteocytes after exposure to HiSMF for 48 h. It was found that the fractal dimension of both microfilament and microtubule of osteocytes was significantly increased under HiSMF (Figure 3C,D). These data indicate that HiSMF promoted the rearrangement of microfilaments and microtubules in osteocytes.

### 3.3. The Effect of HiSMF on the Secretion of Soluble Cytokines and the Expression of Functional Proteins

As the primary coordinator in bone, osteocytes can secrete a variety of soluble molecules that influence the function of other bone cells via gap junctions or paracrine pathways. Several soluble molecules in osteocyte culture supernatants, including ALP, PGE2, NO, and RANKL were tested by commercial kits, separately. The results displayed that treatment of osteocytes with HiSMF for 48 h mildly reduced ALP levels, enhanced the secretion of PGE2 and RANKL, and dramatically decreased NO secretion (Figure 4A).

Two functional proteins of osteocytes, connexin 43 (Cx43) and sclerostin, were examined by Western blot assay. The results showed that Cx43 protein expression was significantly promoted and that sclerostin expression was significantly inhibited in osteocytes that were treated with 16 T magnetic field for 48 h (Figure 4B,C).

### 3.4. The Effect of HiSMF on Cellular Iron Metabolism

In order to elucidate whether the effect of HiSMF on osteocytes was related to iron metabolism, the iron content in osteocytes was examined. The results showed that HiSMF treatment increased the iron content in osteocytes (Figure 5A–C), indicating that the effect of HiSMF on osteocytes may be related to the change of intracellular iron levels.

To clarify the mechanism of the influence of HiSMF on iron content in osteocytes, the expression of iron metabolism-related proteins, including DMT1 (iron absorption), FTH1 (iron storage) and FPN1 (iron export) was determined by Western blot assay. The experimental results indicated that, after exposure under HiSMF, the protein expression of DMT1 increased significantly, while the expression of FPN1 decreased significantly, and HiSMF had no marked influence on FTH1 expression (Figure 5D,E). These results indicated that HiSMF increased the iron content in osteocytes by promoting iron uptake and decreasing iron excretion.

### 3.5. The Effect of Different Levels of Iron on Cell Viability

The cell viability and intracellular iron content of osteocyte increased, and to demonstrate the effect of changes in iron content on osteocyte viability, the effect of different concentrations of FAC and DFO on osteocyte viability was examined. The present results displayed that DFO had no effect on cell viability at low concentrations, but cell viability was inhibited at higher concentrations in a concentration-dependent manner. FAC treatment had no effect on cell viability at low concentrations, promoted cell viability at 0.05 μg/mL iron concentration, but inhibited osteocyte viability at higher iron concentrations (Figure 6). This result suggests that a certain concentration of iron is necessary for osteocyte growth, but too low or too high iron has a toxic effect on osteocytes.

### 3.6. The Effect of Different Levels of Iron on Cell Viability Under HiSMF

To further confirm whether the effect of HiSMF on osteocyte viability was correlated with the alteration of iron levels, the influence of HiSMF combined with different concentrations of DFO or FAC on osteocyte viability was examined. The results indicated that trends in the impact of DFO on cell viability under HiSMF conditions was the same as that of the GMF, but the viability of osteocytes in HiSMF was markedly higher than that in the GMF at all concentrations of DFO. Similarly, the trend of the effect of FAC on cell viability under HiSMF was also the same as that of the control, but only at 0.05 and 0.5 μg/mL iron concentrations. The viability of osteocytes in HiSMF was significantly higher than that of the control (Figure 7). In conclusion, trends in the effects of lower and higher iron on osteocyte viability in HiSMF are consistent with the changes in GMF.

### 3.7. The Effect of Different Levels of Iron in HiSMF on Cell Morphology

Changes in cell morphology were examined in osteocytes exposed to different iron levels under GMF and HiSMF. The results showed no significant changes in cell morphology under different concentrations of DFO treatment, except for DFO at a concentration of 50 μM under HiSMF markedly reduced the number of cell processes. Under FAC treatment conditions, there was no significant change in cell area and perimeter, but the number of dendritic processes of osteocytes under HiSMF was markedly higher than that in the GMF group at iron concentration ≥ 0.5 μg/mL. In addition, the high concentration of iron significantly reduced the length of dendritic processes in osteocytes (Figure 8A,B).

### 3.8. The Effect of Different Levels of Iron in HiSMF on Cell Apoptosis

Changes in cell apoptosis were determined in osteocytes exposed to different iron levels under GMF and HiSMF. The results showed that HiSMF significantly inhibited apoptosis in low DFO levels compared to the GMF, and the inhibition effect was not significant at high concentrations of DFO. Under FAC treatment, the effect of HiSMF on apoptosis was consistent with that of GMF only in the treatment of 50 μg/mL iron, and HiSMF inhibited osteocyte apoptosis at all other iron levels (Figure 9A,B). Moreover, high concentrations of DFO inhibited osteocyte apoptosis, whereas high concentrations of iron promoted cell apoptosis. This result further demonstrates that HiSMF inhibits apoptosis in osteocytes and that cell apoptosis is reduced in iron deficiency and increased in iron overload.

### 3.9. The Effect of Different Levels of Iron in HiSMF on the Cytoskeleton

Changes in the cytoskeleton were evaluated in osteocytes exposed to different concentrations of iron under GMF and HiSMF (Figure 10A). The results showed that under DFO treatment, HiSMF elevated the fractal dimension of both the cellular microfilament and microtubule skeletons compared to the GMF, and 50 μM DFO significantly reduced the fractal dimension of the cytoskeleton in the GMF condition. Under FAC treatment, HiSMF also increased the fractal dimension of both the cellular microfilaments and microtubules compared to the control at low iron concentrations, while no difference was found at high iron concentrations (Figure 10B,C). In addition, morphologically, the rounded osteocytes increased, and the microfilament skeletons were all clustered at the edges of the osteocytes at 50 μg/mL iron concentration treatment. These results further indicate that HiSMF increases the fractal dimension of the microfilament and microtubule skeletons of osteocytes, and the microfilament skeletons rearrange and cluster to the cell edges under iron overload.

### 3.10. The Effect of Different Concentrations of Iron in HiSMF on the Secretion of Soluble Cytokines

The secretion of soluble cytokines by osteocytes was examined at high and low concentrations of iron under GMF and HiSMF. The present results show that HiSMF significantly inhibited the secretion of ALP and NO and facilitated the secretion of PGE2 and RANKL at different concentrations of DFO or FAC treatment. DFO inhibited the concentration of ALP secreted by osteocytes under GMF and HiSMF. Low levels of iron promote ALP secretion, while high concentrations inhibit its secretion (Figure 11A). The secretion of PGE2 was notably elevated by DFO under GMF and HiSMF, and different levels of iron increased PGE2 content under HiSMF (Figure 11B). The secretion of NO was significantly stimulated by different concentrations of DFO or FAC under GMF and HiSMF condition (Figure 11C). Low concentrations of DFO could markedly promoted RANKL secretion, and different levels of iron all promoted the secretion of RANKL by osteocytes under GMF and HiSMF conditions (Figure 11D).

## 4. Discussion

Most studies on the regulation of bone cells by magnetic fields have focused on osteoblasts and osteoclasts, while there are few reports on the biological effects of osteocytes under magnetic fields. Only some preliminary studies of how a 16 T magnetic field generated by a large gradient superconducting magnet affects osteocytes has been conducted in our laboratory previously. The osteocyte line MLO-Y4 was cultured in 16 T for 24 and 48 h, and the viability of osteocytes was not affected significantly [7]. However, in the current study, the viability of osteocytes cultured for 48 h under HiSMF was enhanced. The possible reason for this inconsistent result is that we have recently improved the cell culture environment for HiSMF of 16 T [20]. Morphologically, and consistent with previous findings, the number of osteocyte processes as well as the spreading area of cells did not change significantly under HiSMF compared to the GMF group. In addition, our study also showed that HiSMF significantly reduced apoptosis in osteocytes, suggesting that the promotion of osteocyte viability by HiSMF may be related to a reduction in apoptosis.

The cytoskeleton, which fills the entire cytoplasm from the nucleus to the plasma membrane, is a complex network of interconnected microfilaments and microtubules [22]. The cytoskeleton not only plays a crucial role in supporting cellular morphology, standing against external forces, and maintaining the orderliness of the intracellular structure, but is also involved in many cellular activities [23]. The intricacy and irregularity of the cytoskeleton make it difficult to be analyzed quantitatively; thus, previous researchers have described the structurally altered cytoskeleton only as a rearrangement of the skeleton [24]. Afterward, the application of fractal analysis within the field of biology has made it possible to describe such complex structures from the impossible to the possible [25]. The most important feature of fractal theory is that it goes beyond the traditional barriers of one-dimensional lines, two-dimensional surfaces, three-dimensional planes, and four-dimensional space-time, and is more closely aligned with the characterization of the real properties and states in complex systems and more in keeping with the multiplicity and complicacy of objective things. The fractal dimension is the most important covariate to describe fractals. In biological studies, fractal dimensions have been used to describe the changes of actin and tubulin skeletons [26,27]. In the present study, the fractal dimension was also used to describe the effects of HiSMF and different concentrations of DFO and FAC treatments on the microfilament and microtubule skeletons of MLO-Y4 cells. The results displayed that HiSMF markedly increased the fractal dimension of microfilaments and microtubules, indicating that HiSMF promoted the skeletal arrangement of osteocytes. Under the GMF, a high concentration of DFO treatment significantly decreased the fractal dimension of the cytoskeleton, indicating that cytoskeleton rearrangement and structural disruption occurred in the absence of cellular iron. The fractal dimension of the microfilament skeleton of osteocytes was significantly reduced under high concentrations of FAC treatment, but there was no effect on the microtubule skeleton, indicating that iron overload can disrupt the microfilament skeleton of osteocytes. In addition, morphologically, the microfilament skeleton of osteocytes was clustered at the edges of the cells when treated with 50 μg/mL iron, indicating that high concentrations of iron resulted in an altered distribution of the microfilament skeleton in osteocytes.

In bone remodeling, osteocytes can secrete a variety of cytokines to regulate the differentiation and function of osteoclasts and osteoblasts, thereby modulating bone homeostasis [28]. Sclerostin, a glycoprotein specifically secreted by osteocytes, is regulated by the *SOST* gene and is an antagonist of the bone morphogenetic protein (BMP) family that regulates bone formation [29]; it also inhibits the Wnt signaling pathway, thereby inhibiting bone formation [30]. The results of our study showed that 16 T HiSMF significantly inhibited the protein levels of sclerostin in osteocytes. In addition, HiSMF mildly promoted the secretion of PGE2 and RANKL and dramatically blocked the secretion of NO. PGE2 plays a vital role in the process of bone remodeling, with low concentrations stimulating bone formation and increasing bone mass, and high concentrations stimulating bone resorption [31]. In contrast to the role of PGE2, low levels of NO enhanced osteoclast formation and the associated bone resorption, and high levels accelerated the proliferation and differentiation of osteoblasts [32]. RANKL is an essential cytokine for osteoclastic survival, proliferation and differentiation [33]. Moreover, osteocytes abundantly express Cx43 proteins, which form both gap junctions and hemichannels. Cx43 gap junctions can facilitate the passage of small signaling molecules between coupled cells, whereas Cx43 hemichannels can mediate communication between the cell and the extracellular environment [34]. Herein, we found that HiSMF can promote the expression of Cx43 protein in osteocytes. Therefore, 16 T HiSMF modulates cytokine secretion by osteocytes and promotes their communication with adjacent cells, which may affect the balance of bone remodeling; further studies are needed to investigate.

Since 1984, when Vernejoul et al. [35] performed intramuscular dextran iron to pigs, causing an iron overload and finding that iron overload can affect bone remodeling, studies related to iron and bone metabolism have increased. As the three major cells in bone— osteoclasts, osteoblasts, and osteocytes—there have been many studies related to iron and osteoblasts or osteoclasts [36], but few studies related to iron and osteocytes. We recently demonstrated that iron overload inhibited cell viability, caused apoptosis and increased RANKL expression and secretion in osteocytes [37]. Consistent with these results, the present study also found that high concentrations of FAC significantly inhibited cell viability, caused apoptosis, and stimulated RANKL secretion in MLO-Y4 cells. In addition, our study indicated that too-low concentrations of iron also decreased osteocyte viability but reduced osteocyte apoptosis. Thus, DFO reduces osteocyte viability, not through the apoptotic pathway, but maybe via inhibiting cell proliferation, leading to other modes of cell death. 

Osteocytes, cells with a characteristic morphology, have multiple long, neuron-like dendritic processes throughout the lacunar canalicular system in bone [38]. The dendritic network structure of osteocytes confers its mechanical sensitivity and allows it to extensively communicate with adjacent cells on bone surfaces, including with osteoblasts and osteoclasts [39]. Defects in the dendrite network of osteocytes may induce skeletal fragility in aging and glucocorticoid-induced osteoporosis [40,41]. In the present study, although high concentrations of iron elevated the number of dendritic processes, it decreased dendrite length in osteocytes. Morphologically, it was observed that high iron levels led to an increase in rounded osteocytes with many short processes. Consequently, it is possible that iron overload can promote dendrite formation but inhibit the extension of dendrites. However, this conclusion has yet to be verified by in vivo studies, and the mechanisms involved need to be explored in further studies.

## 5. Conclusions

In the current study, we first systematically explored the influence of HiSMF on the biological behavior of osteocytes using a 16 T magnetic field. The results showed that HiSMF promoted cellular viability, decreased apoptosis, and increased the fractal dimension of the cytoskeleton in osteocytes. The possible mechanism was that the HiSMF of 16 T promoted the expression of DMT1 and inhibited FPN1 expression in osteocytes, such that iron levels in osteocytes were increased. Assays of functional protein expression in osteocytes revealed that HiSMF promoted Cx43 protein expression and inhibited the expression of sclerostin. Assays of soluble cytokines secreted by osteocytes indicated that HiSMF decreased the secretion of ALP and NO and increased the secretion of PGE2 and RANKL by osteocytes. Then, the current study investigated the combined effect of HiSMF and exogenous DFO or FAC on the biological behavior of osteocytes. It was found that under the condition of DFO or FAC treatment, HiSMF still promoted cell viability, reduced apoptosis, and promoted cytoskeletal rearrangement in osteocytes. In conclusion, our results suggest that the modulation of osteocytic iron metabolism by HiSMF may be involved in the altered biological effects of osteocytes under HiSMF exposure. These data enrich the biological effects under HiSMF and provide a theoretical basis for the development of magnetic field machines for the treatment of bone-related diseases. However, further in vitro and in vivo studies are needed to determine whether HiSMF affects osteocyte function by modulating cellular iron metabolism.

## Figures and Tables

**Figure 1 cells-10-03519-f001:**
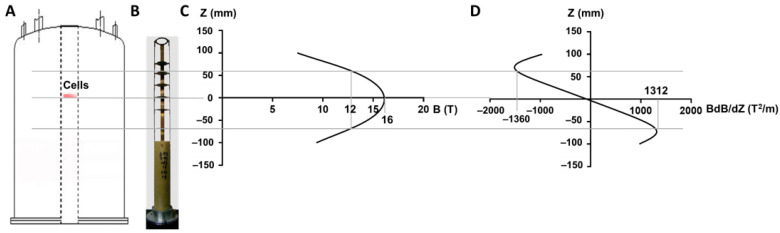
Abbreviated drawing of the magnetic field exposure systems. (**A**) The location of cell culture is shown in the superconducting magnet. (**B**) The object stage of cell culture. (**C**) The corresponding magnetic field intensity for the different positions. (**D**) The magnetic field gradients corresponding to different positions.

**Figure 2 cells-10-03519-f002:**
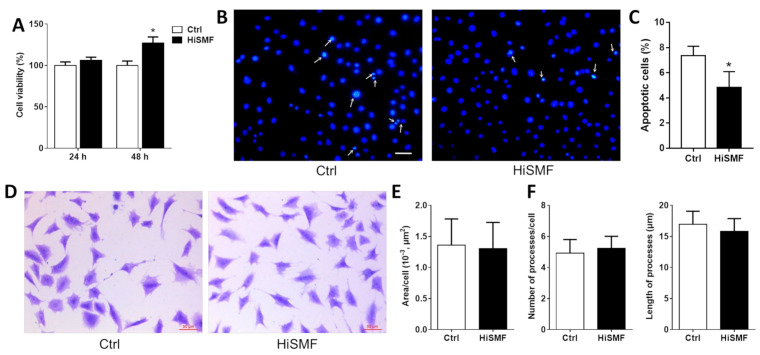
The influence of high static magnetic fields (HiSMF) on cell growth in osteocytes. (**A**) Cell viability in osteocytes was detected by CCK-8 after exposure to HiSMF for 48 h. (**B**) Apoptotic cells were detected by Hoechst staining. (**C**) Statistics on the proportion of apoptotic cells. (**D**) Observation of cellular morphology by crystal violet staining. (**E**) Analysis of the cellular area using Image-Pro Plus 6.0 software (Media Cybernetics, Houston, TX, USA). (**F**) Analysis of the amount and length of dendritic processes in osteocytes. *n* ≥ 3. Data shown as mean ± SD. * *p* < 0.05.

**Figure 3 cells-10-03519-f003:**
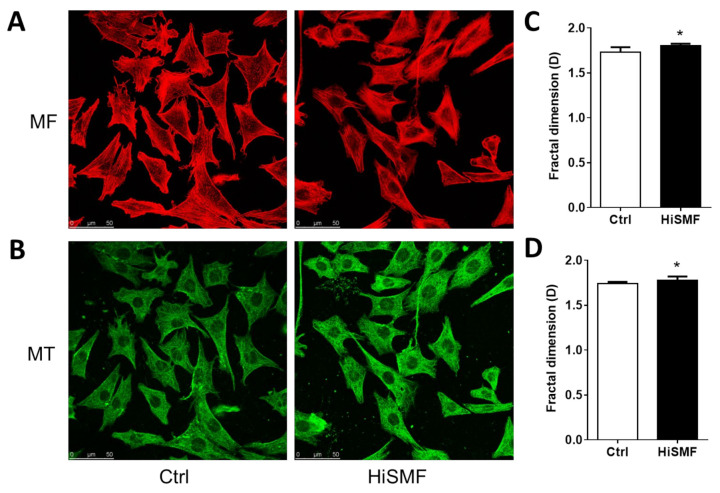
The influence of HiSMF on the cytoskeleton in osteocytes. (**A**) Fluorescent staining of the microfilament (MF) in osteocytes using a rhodamine-labeled phalloidin. (**B**) Fluorescent staining of microtubules (MT) in osteocytes with anti-tubulin antibody and FITC-labeled secondary antibody. (**C**) Analysis of the fractal dimension of microfilaments using ImageJ software. (**D**) Analysis of fractal dimensions of microtubules using ImageJ 1.8.0 software (NIH, Bethesda, MD, USA). *n* = 50. Data shown as mean ± SD. * *p* < 0.05.

**Figure 4 cells-10-03519-f004:**
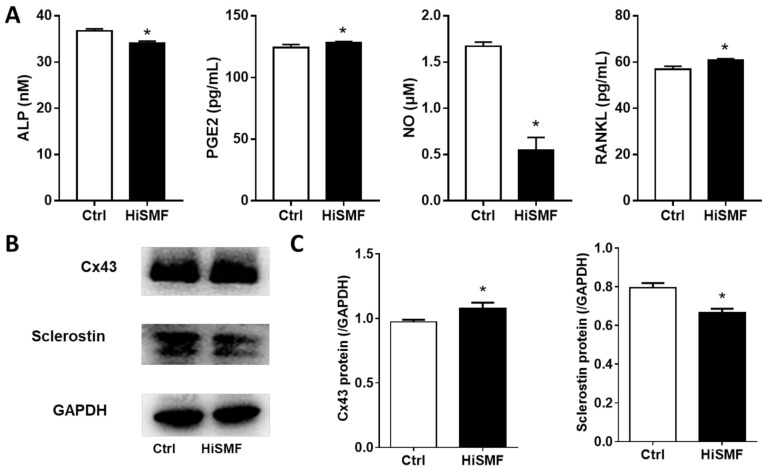
The effect of HiSMF on the secretion of soluble cytokines and the expression of functional proteins. (**A**) Several soluble molecules in osteocyte culture supernatants, including alkaline phosphatase (ALP), prostaglandin E2 (PGE2), nitric oxide (NO), and Nuclear Factor κB Receptor Activator Ligand (RANKL) were examined by commercial kits. (**B**) Imaging of proteins detected by Western blot and imaged via a ECL Plus Western Blotting Detection System (Tanon, Shanghai, China). (**C**) The immunoreactive grayscale band was determined using ImageJ software. *n* = 5. Data shown as mean ± SD. * *p* < 0.05.

**Figure 5 cells-10-03519-f005:**
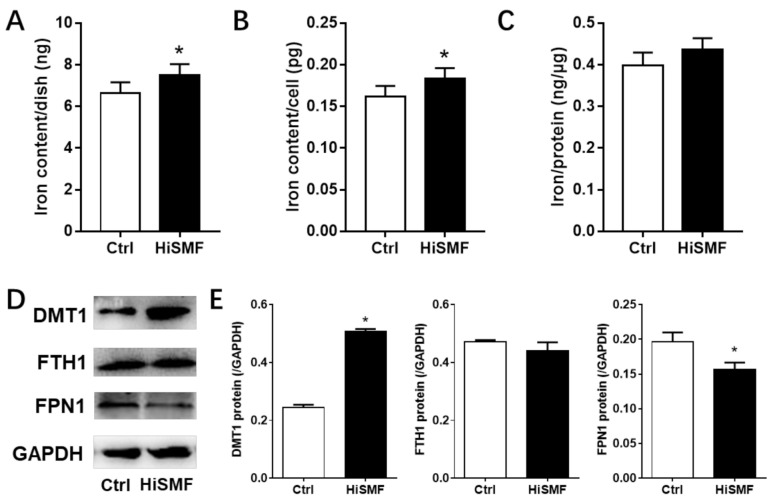
The influence of HiSMF on cellular iron metabolism in osteocytes. (**A**) Iron levels in osteocytes was examined by atomic absorption spectroscopy (AAS) and displayed as milligram (mg) per petri dish. (**B**) Total iron levels in each petri dish were normalized by the total number of cells. (**C**) Total iron levels in each petri dish were normalized by the content of total protein. (**D**) Images of proteins detected by Western blot were photographed by an ECL Plus Western Blotting Detection System. (**E**) The immunoreactive grayscale band was determined using ImageJ software. *n* = 5. Data shown as mean ± SD. * *p* < 0.05.

**Figure 6 cells-10-03519-f006:**
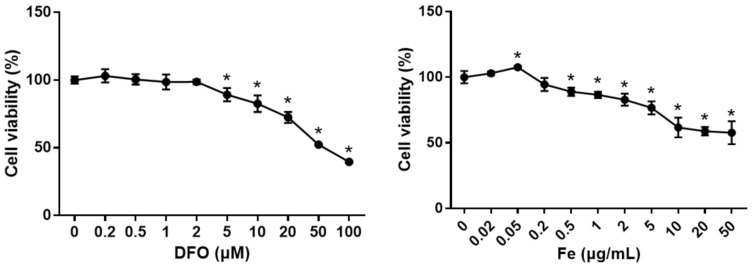
Cellular viability of osteocytes was measured by CCK-8 in management with different concentrations of deferoxamine (DFO) and ferric ammonium citrate (FAC; convert into Fe levels) at 48 h. *n* = 5. Data shown as mean ± SD. * *p* < 0.05.

**Figure 7 cells-10-03519-f007:**
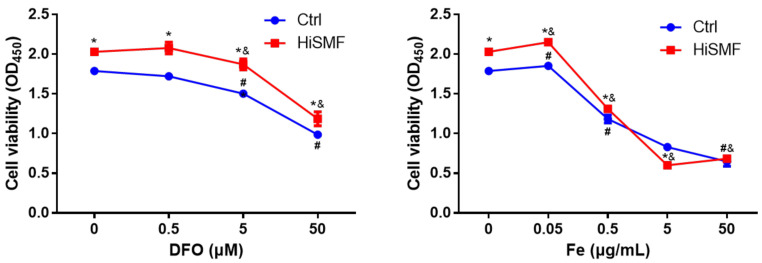
Effect of the combination of HiSMF and DFO or FAC on cell viability in osteocytes. *n* = 5. Data shown as mean ± SD. * *p* < 0.05 vs. Ctrl, ^#^
*p* < 0.05 vs. Ctrl-0 (0 μM DFO or 0 μg/mL Fe in the Ctrl group), ^&^
*p* < 0.05 vs. HiSMF-0 (0 μM DFO or 0 μg/mL Fe in the HiSMF group).

**Figure 8 cells-10-03519-f008:**
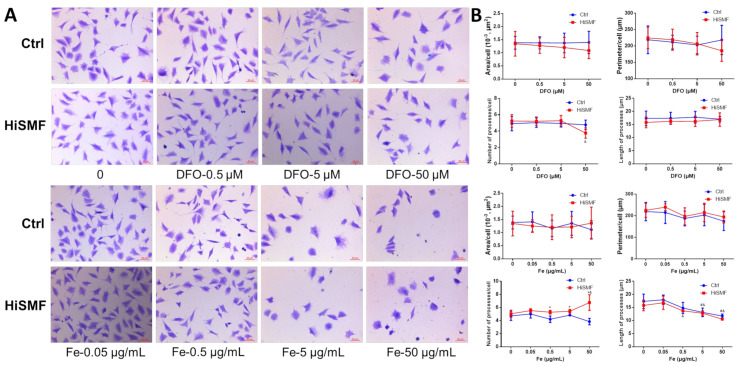
Effect of the combination of HiSMF and DFO or FAC on cellular morphology in osteocytes. (**A**) Observation of cellular morphology by crystal violet staining. (**B**) Analysis of morphological parameters of osteocytes using Image-Pro Plus software. *n* = 50. Data shown are mean ± SD. * *p* < 0.05 vs. Ctrl, ^#^
*p* < 0.05 vs. Ctrl-0 (0 μM DFO or 0 μg/mL Fe in the Ctrl group), ^&^
*p* < 0.05 vs. HiSMF-0 (0 μM DFO or 0 μg/mL Fe in the HiSMF group).

**Figure 9 cells-10-03519-f009:**
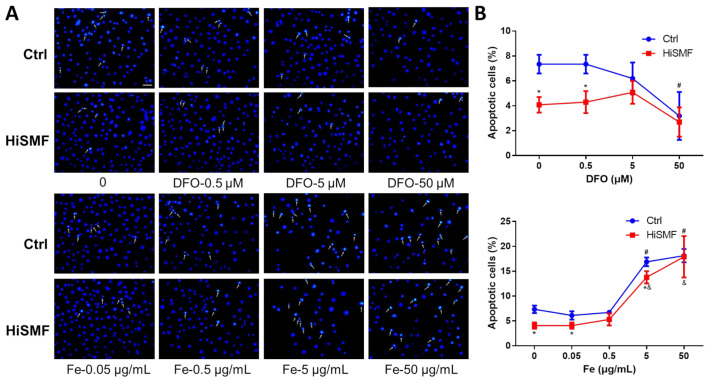
Effect of the combination of HiSMF and DFO or FAC on cell apoptosis in osteocytes. (**A**) Apoptotic cells were detected by Hoechst staining. (**B**) Statistics on the proportion of apoptotic cells. *n* = 5. Data shown are mean ± SD. * *p* < 0.05 vs. Ctrl, ^#^
*p* < 0.05 vs. Ctrl-0 (0 μM DFO or 0 μg/mL Fe in the Ctrl group), ^&^
*p* < 0.05 vs. HiSMF-0 (0 μM DFO or 0 μg/mL Fe in the HiSMF group).

**Figure 10 cells-10-03519-f010:**
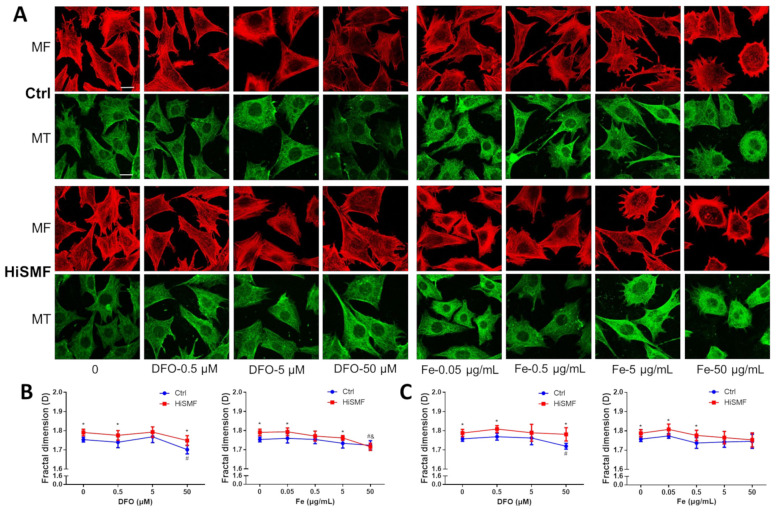
Effect of the combination of HiSMF and DFO or FAC on the cytoskeleton in osteocytes. (**A**) Fluorescent staining of the microfilament in osteocytes using a rhodamine-labeled phalloidin and fluorescent staining of microtubules in osteocytes with anti-tubulin antibody and FITC-labeled secondary antibody. Bar = 25 μm. (**B**) Analysis of the fractal dimension of microfilaments using ImageJ software. (**C**) Analysis of fractal dimensions of microtubules using ImageJ software. *n* = 5. Data shown are mean ± SD. * *p* < 0.05 vs. Ctrl, ^#^
*p* < 0.05 vs. Ctrl-0 (0 μM DFO or 0 μg/mL Fe in the Ctrl group), ^&^
*p* < 0.05 vs. HiSMF-0 (0 μM DFO or 0 μg/mL Fe in the HiSMF group).

**Figure 11 cells-10-03519-f011:**
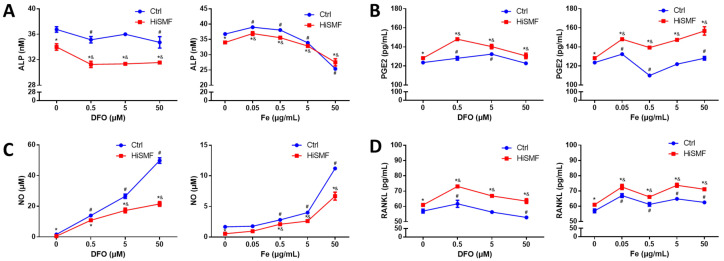
The influence of the combination of HiSMF and DFO or FAC on soluble molecules secreted from osteocytes, including: (**A**) ALP, (**B**) PGE2, (**C**) NO, and (**D**) RANKL. *n* = 5. Data shown are mean ± SD. * *p* < 0.05 vs. Ctrl, ^#^
*p* < 0.05 vs. Ctrl-0 (0 μM DFO or 0 μg/mL Fe in the Ctrl group), ^&^
*p* < 0.05 vs. HiSMF-0 (0 μM DFO or 0 μg/mL Fe in the HiSMF group).

## Data Availability

The datasets in this study are available from the corresponding author upon reasonable request.

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
