# Peer review of "Effect of High Static Magnetic Fields on Biological Activities and Iron Metabolism in MLO-Y4 Osteocyte-like Cells"

_cells, 2021, doi:10.3390/cells10123519_

Round 1
Reviewer 1 Report
Comments and Suggestions for Authors
- The authors addressed the reader to reference 16 for the description of used apparatus. They should provide a complete description of their experimental platform as well as a picture of the setup of the experiment showing object stage. They also used three different magnetic field gradients which seems that the intensity of magnetic field was position-dependent. How they become sure about the homogeneity of generated field during the treatment of cells?
- Osteocytes are cells which come from osteoblasts. These cells do not divide due to reside within bone matrix and have a long life time, while MLO-Y4 cells are a cell-line. How do the authors generalize the results of this study to Osteocytes cells?
- I suggest the authors to provide cell growth curve for the cell line. Please prepare more explanation indicating, the cells in which part of growth curve and which subculture were treated.
- The authors have used the word Cell Viability in the title of the method and in its description cell activity, “Then cell activity of MLO-Y4 cells was determined by the CCK-8 method using the...”.If these two words have the same meaning, I will therefore suggest to use cell viability in the whole of the text.
- The description for applied method for cell apoptosis assay is not clear and need more explanation. For example how many times did you count the cells?
- In the result section, the title “The Effect of HiSMF on Cell Growth “has been repeated twice. I will suggest either merge the two parts or change the titles.
- According to the results, did the magnetic field increase cell viability by reducing apoptosis or by increasing cell division? Please provide more explain about it.
Author Response
Point 1: The authors addressed the reader to reference 16 for the description of used apparatus. They should provide a complete description of their experimental platform as well as a picture of the setup of the experiment showing object stage. They also used three different magnetic field gradients which seems that the intensity of magnetic field was position-dependent. How they become sure about the homogeneity of generated field during the treatment of cells? 

Response 1: We have added a complete description of our experimental platform in the manuscript and supplemented a schematic diagram (Figure1) and showed the object stage in a picture. Maybe you misunderstood, we only used one position in this study, that is the position of 16 T with 0 T2/m.
Point 2: Osteocytes are cells which come from osteoblasts. These cells do not divide due to reside within bone matrix and have a long life time, while MLO-Y4 cells are a cell-line. How do the authors generalize the results of this study to Osteocytes cells?
Response 2: MLO-Y4 cells is the first osteocyte-like cell line developed by Prof. Lynda F. Bonewald (please find the reference “Kato Y, Windle JJ, Koop BA, Mundy GR, Bonewald LF. Establishment of an osteocyte-like cell line, MLO-Y4. J Bone Miner Res. 1997 Dec;12(12):2014-23. doi: 10.1359/jbmr.1997.12.12.2014.”), it is a murine osteocyte-like cell line, has been used extensively to investigate osteocyte function.
Point 3: I suggest the authors to provide cell growth curve for the cell line. Please prepare more explanation indicating, the cells in which part of growth curve and which subculture were treated.
Response 3: The cell growth curve of MLO-Y4 cells can be find in the reference “Kato Y, Windle JJ, Koop BA, Mundy GR, Bonewald LF. Establishment of an osteocyte-like cell line, MLO-Y4. J Bone Miner Res. 1997 Dec;12(12):2014-23. doi: 10.1359/jbmr.1997.12.12.2014.”. MLO-Y4 cells at the logarithmic phase were seeded and treated in our study. We have described this in the methods.
Point 4: The authors have used the word Cell Viability in the title of the method and in its description cell activity, “Then cell activity of MLO-Y4 cells was determined by the CCK-8 method using the...”.If these two words have the same meaning, I will therefore suggest to use cell viability in the whole of the text.
Response 4: We have changed “cell activity” to “cell viability” in the whole of the manuscript.
Point 5: The description for applied method for cell apoptosis assay is not clear and need more explanation. For example how many times did you count the cells?
Response 5: We have added more description for cell apoptosis assay in the methods section. “The total number of cell nucleus as well as the number of apoptotic nucleus in each sight were counted using Image-Pro Plus software. Three sights per sample were counted. The percentage of apoptotic cells were determined by calculating the ratio of the number of apoptotic nucleus to the total number of cell nucleus.”
Point 6: In the result section, the title “The Effect of HiSMF on Cell Growth “has been repeated twice. I will suggest either merge the two parts or change the titles.
Response 6: It is our negligence. We have changed the tile of the result section 3.2 to “The Effect of HiSMF on Cytoskeleton”
Point 7: According to the results, did the magnetic field increase cell viability by reducing apoptosis or by increasing cell division? Please provide more explain about it.
Response 7: Although our results demonstrated that magnetic fields reduce apoptosis, it do not prove that this is the only reason why magnetic fields increase cell viability. Further experiments are needed to demonstrate the effect of magnetic fields on cell division.

Reviewer 2 Report
Comments and Suggestions for Authors
The authors should underline in the introduction that they examined the influence of HiSMF on mouse cell line MLO-Y4 which stems out of transgenic mouse expressing SV40 large T-antigen.
In methods they didn't mention about treatment of cells with DFO and FAC while decribing in results.
The abbreviation like DFO, FAC and others should be Unfolded in text where first mentioned.
The conclusion would be more cautious about the influence of iron metabolism on changes of osteocytes made by HiSMF.
Author Response
Point 1: The authors should underline in the introduction that they examined the influence of HiSMF on mouse cell line MLO-Y4 which stems out of transgenic mouse expressing SV40 large T-antigen.

Response 1: We have added a description for the source of cell line MLO-Y4 in the introduction.
Point 2: In methods they didn't mention about treatment of cells with DFO and FAC while decribing in results.
Response 2: We have added a description of treatment of cells with DFO and FAC in methods.
Point 3: The abbreviation like DFO, FAC and others should be Unfolded in text where first mentioned.
Response 3: We have examined all the text in the manuscript, and unfolded the abbreviation in text where first mentioned.
Point 4: The conclusion would be more cautious about the influence of iron metabolism on changes of osteocytes made by HiSMF.
Response 4: Yes, we've been cautious, so we've used the word "may" to indicate the uncertainty.

Reviewer 3 Report
Comments and Suggestions for Authors
This manuscript by Yang et al, investigates the effect of high static magnetic fields on osteocytes and the role of iron in these processes. The authors report that HiSMF promote cellular viability, reduce apoptosis, affect rearrangement of the cytoskeleton and regulate the release of bone-related proteins. Importantly, the authors found that the actual concentration or iron can have differential effect on osteocytes under HiSMF conditions. For this work, they used a variety of approaches and methods.
The work is very interested, well presented, with novel findings. The authors should address the points below.
- The importance of investigating the effects of HiSMF is not mentioned at all. Please provide some information.
- In the conclusions, the authors should include the contribution/importance of this work to the field and what could be the potential next steps.
- The manuscript needs careful editing. For example, line 25 – finds, line 69 – skeleton, line 106 – crystalline, line 373 – osteoblasts, line 402 - increasingly etc
- The term “dendrites” must be replaced by “processes” or “long dendritic processes” in the text and in Figures/legends. This is the official term.
- All bar charts should be replaced with scatter plots since the number of samples (or replicates) is low, e.g. n-5.
- For the determination of ALP, do the authors mean that they determined activity? In Figure 2, the ALP units reflect concentration. It would be more suitable to measure the enzyme activity; however, it is known that osteocytes produce a very small amount of ALP and this measurement could be skipped as does not provide important information for this work.
- It would be more appropriate to measure the levels of secreted sclerostin in the culture supernatant, and not by lysing the cells, by ELISA rather than Western blot.
- In Results 3.6, how osteocyte activity was measured? Do the authors mean osteocyte viability as shown in Fig6? If not, a definition of activity must be provided.
- The cell treatments with DFO and FAC are not included in the Methods. Please provide a detailed description.
- Line 386, replace expression with protein levels/abundance.
Author Response
Point 1: The importance of investigating the effects of HiSMF is not mentioned at all. Please provide some information.

Response 1: We have added a paragraph on the importance of investigating the effects of HiSMF in the introduction.
Point 2: In the conclusions, the authors should include the contribution/importance of this work to the field and what could be the potential next steps.
Response 2: We have added a description as follow “These data enrich the biological effects under HiSMF and provide a theoretical basis for the development of magnetic field devices for the treatment of bone-related diseases. However, further in vivo and in vitro studies are needed to determine whether HiSMF affect osteocyte function by modulating cellular iron metabolism”
Point 3: The manuscript needs careful editing. For example, line 25 – finds, line 69 – skeleton, line 106 – crystalline, line 373 – osteoblasts, line 402 - increasingly etc.
Response 3: We have double-checked the whole text and corrected some spelling and grammatical errors, including you listed.
Point 4: The term “dendrites” must be replaced by “processes” or “long dendritic processes” in the text and in Figures/legends. This is the official term.
Response 4: We've replaced the term “dendrites” with “processes” or “dendritic processes” in the text and in Figures/legends.
Point 5: All bar charts should be replaced with scatter plots since the number of samples (or replicates) is low, e.g. n-5.
Response 5: We believe that cellular experiments with 5 replicate samples are not low, and it is enough for difference statistics, and many published studies have only 3 replicates, so scatter plots are not necessary.
Point 6: For the determination of ALP, do the authors mean that they determined activity? In Figure 2, the ALP units reflect concentration. It would be more suitable to measure the enzyme activity; however, it is known that osteocytes produce a very small amount of ALP and this measurement could be skipped as does not provide important information for this work.
Response 6: Since we are evaluating the secretory capacity of osteocytes in this study, thus we determined the concentration of ALP rather than the enzyme activity. It is true that osteocytes produce a very small amount of ALP, but we examined ALP levels aim to evaluate the secretory capacity of osteocytes.
Point 7: It would be more appropriate to measure the levels of secreted sclerostin in the culture supernatant, and not by lysing the cells, by ELISA rather than Western blot.
Response 7: For secreted sclerostin, which is also generated by osteocytes through a transcriptional-translational process, it is also appropriate to determine protein expression by Western blot.
Point 8: In Results 3.6, how osteocyte activity was measured? Do the authors mean osteocyte viability as shown in Fig6? If not, a definition of activity must be provided.
Response 8: Yes, osteocyte activity means osteocyte viability in our study, we have changed “activity” to “viability” in the whole of the manuscript.
Point 9: The cell treatments with DFO and FAC are not included in the Methods. Please provide a detailed description.
Response 9: We have added a detailed description of treatment of cells with DFO and FAC in methods.
Point 10: Line 386, replace expression with protein levels/abundance.
Response 10: We have replaced “expression” with “protein levels”.
